

# Participation in youth sports influences sarcopenia parameters in older adults

Kaja Teraž[1,2], Miloš Kalc[2,3], Boštjan Šimunič[2], Uros Marusic[2,4], Primož Pori[1], Saša Pišot[2] and Rado Pišot[2]

[1] Faculty of Sport, University of Ljubljana, Ljubljana, Slovenia
[2] Institute for Kinesiology Research, Science and Research Centre Koper, Koper, Slovenia
[3] Institute of Sports Medicine, Faculty of Medicine, University of Maribor, Maribor, Slovenia
[4] Department of Health Sciences, Alma Mater Europaea - ECM, Maribor, Slovenia

Corresponding author
Kaja Teraž, kaja.teraz@zrs-kp.si

## ABSTRACT

**Background:** The degree of deterioration in sarcopenia parameters may be affected by a person's level of physical activity (PA) and sedentary behavior (SB). Our study focused on examining the PA and SB of active older adults including those with and without history of sports in youth.

**Methods:** Forty-four participants (20 men and 24 women, mean age of total sample $76.1 \pm 5.2$ years) were included in analysis of PA, SB habits and sarcopenia parameters, determined by skeletal muscle index, hand-grip strength, gait speed, Timed Up and Go tests (TUG). PA and SB were recorded with accelerometers. Our primary aim was to compare participants with (AH) or without a sport history in youth (NAH), in their sarcopenia parameters and PA and SB habits.

**Results:** When divided participants in two groups (AH and NAH) and adjusting for age, we have detected the differences for skeletal muscle index ($p = 0.007$) and hand-grip strength ($p = 0.004$) in favor of participants who were engaged in sports in youth. We did not find any differences in PA and SB habits between the AH and NAH groups. After adjusting for age, participants with a higher number of daily steps, longer moderate to vigorous physical activity (MVPA) bouts, a higher number of MVPA bouts in a day and higher overall MVPA engagement achieved better results in hand-grip strength and TUG. Participants with lower SB had better TUG and gait speed results.

**Conclusions:** Our findings suggest that engaging in sports activities in youth can make a difference with sarcopenia parameters. Although we found no differences in PA and SB habits between participants with AH and NAH, participants with an athlete history performed better results in sarcopenia parameters.

## INTRODUCTION

Physical activity is known to have numerous health benefits and is a key strategy to prevent age-related deterioration of body systems functioning. It is especially important for slowing the age-related loss of muscle mass and strength (*Meier & Lee, 2020*). An active lifestyle is presented in the literature as an ideal type of healthy lifestyle. Physical activity (PA) has both preventive and therapeutic health effects (*Sudeck et al., 2021*) and can have a

inhibitory effect for many chronic non-communicable diseases (NCDs) or disorders (*Pedersen & Saltin, 2015*; Physical Activity Guidelines Advisory Committee; *PAGAC, 2018*). One of the disorders that occur in older age is also sarcopenia. In addition, participation in sports during youth may positively impact older adults in several ways; studies suggest that exercise habits developed and practiced in young adulthood may lead to healthier aging (*Cruz-Jentoft et al., 2019*; *Sayer et al., 2008*). In this regard, it remains unclear whether exercise habits in adolescence contribute to the prevention of sarcopenia in old age.

Sarcopenia is a generalized skeletal muscle disorder (*Cruz-Jentoft et al., 2019*) which is more common among older adults. It is categorized as loss of muscle strength, muscle mass and physical performance (*Cruz-Jentoft et al., 2019*). The decrease in muscle mass and strength with age is influenced by several factors, including health status, genetic predisposition, level of physical activity, and participation in strength and power training. Adequate nutrition and the starting level of muscle mass in early adulthood also play a role (*Keller, 2019*; *Keller & Engelhardt, 2013*). Sarcopenia is highly associated with detrimental clinical outcomes such as falls, physical disability, fractures, cognitive impairment, hospitalization and therefore all-cause mortality (*Cruz-Jentoft et al., 2019*; *Van Ancum et al., 2020*), but can be controlled with behavioral factors such as PA.

Low levels of PA have already been shown to be a major factor in the development of sarcopenia (*Dent et al., 2018*; *Lee et al., 2018*; *Steffl et al., 2017*). For example, a low number of daily steps may promote progressive loss of muscle strength and mass (*Bell et al., 2016*; *Woo, 2017*). On the other hand, increased PA is related with lower odds of sarcopenia parameters (*Sánchez-Sánchez et al., 2019*; *Westbury et al., 2018*). For example, even an additional hour of light intensity physical activity per week, can lead to a better result on a physical performance test (*Scott et al., 2021*). It was also established, that older adults who developed sarcopenia, spent less time in moderate to vigorous physical activity (MVPA) in comparison to older adults without sarcopenia (*Scott et al., 2021*). Older adults with high levels of MVPA tend to have higher baseline values for sarcopenia parameters such as muscle strength, muscle mass, and gait speed (*Mijnarends et al., 2016*). However, older adults who are overall physically active can still be highly sedentary during the day (*Barone Gibbs et al., 2017*). SB can accelerate the loss of muscle quality and quantity (*Shad et al., 2016*) and has an overall negative impact on human health (*Rezende et al., 2014*). *Smith et al. (2019)* stated that an additional hour of sitting in a day may increase odds for sarcopenia. Moreover, higher sedentary time may increase risk for sarcopenia independent of time spent in MVPA (*Aggio et al., 2016*). The latest version of the World Health Organization guidelines (WHO) on PA and SB states that older adults should engage in varied physical activities at least three times per week, including functional balance and strength training, and limit the amount of time spent sitting (*Bull et al., 2020*). Moreover, there is only few guidelines (for specific countries such as USA and UK) that were updated and recommend that older adults should not spend long periods in SB (*GOV.UK, 2019*; *Piercy et al., 2018*), but no concrete recommendations can be found in the literature. The

literature estimates that approximately 67% of older adults spend more than 8.5 h per day sedentary (*Harvey, Chastin & Skelton, 2013*), but replacing 1 h of sedentary time with 1 h of physical activity per day could significantly improve outcomes relevant to sarcopenia (*Sánchez-Sánchez et al., 2019*). Physical performance seemed to be affected by the total time spent in sedentary behavior since most of the studies displayed a significant relationship (*Eberl, 2021*). Nevertheless, the literature already suggested that in addition to total SB time, it is important to avoid prolonged uninterrupted periods of SB time (*Healy et al., 2008*). In the literature, little is known about the active older adults and the association with sarcopenia. Moreover, there is still little known about active older adults that are usually engaged in regular PA and about their habits of sedentarism and SB.

It has been suggested that exercise habits developed and practiced at young age and early adulthood can contribute to a healthier aging process (*Cruz-Jentoft et al., 2019*; *Sayer et al., 2008*). Therefore, we were interested in active older adults who were engaged in sport history in youth and their sarcopenia parameters. We hypothesized that there will be differences in sarcopenia parameters between participants who were engaged in organized forms of sport in their youth and participants who have no history of sports participation in youth. We were also interested in the habits of physical activity and sedentary behavior between participants with sports history in youth and participants without sports history in youth. Moreover, as far as we know, there was no study that examined the effect of total SB time or number and length of SB bouts of active older adults and sarcopenia parameters among active older adults. Second aim was to evaluate the different SB and PA patterns in active older adults and to evaluate their sarcopenia parameters.

## MATERIALS AND METHODS

### Participants

This cross-sectional observational study examines a group of older adults aged 65 or older who were previously involved in the Physical Activity and Nutrition for Great Aging (PANGeA) mass measurements study in 2013. This study included 52 active, older adults, aged between 65 and 85 years (22 men and 30 women) assessed at a follow-up PANGeA mass measurement. Participants were invited to the measurements *via* mail and additionally *via* phone call. Measurements included anthropometric and body composition measurements, physical performance tests, objectively measured PA and SB with accelerometers and questionnaire with information about participation in youth sports, socio-demographic data, information on health status, medications. Data were collected as previously described in *Teraž et al. (2023)*.

The study was conducted in accordance with the Declaration of Helsinki and ap-proved by the National Ethical Committee of Slovenian Ministry of Health (ethical approval no. 0120-76/2021/6) and confirmed by the ZRS Koper Scientific Council no. 0624-77/21. Moreover, the clinical trial protocol was registered on ClinicalTrials.gov, Identifier: NCT04899531. Written informed consent was obtained from all participants involved in the study.

## Measurements

### Anthropometric characteristics and body composition

Body mass was determined with a precision of 0.1 kilogram using a manual weighing scale (Seca 709; Seca, Hamburg, Germany) with the participant wearing only in light underwear and no shoes. Body height was measured with a precision 0.5 centimeters using a standardized wall-mounted height board. Body mass index was calculated as body mass divided by body height squared.

For the assessment of body composition, tetrapolar bioimpedance (BIA 101 Anniversary; Akern-Srl, Florence, Italy) was employed after participants had been in a supine position for 30 min. A muscle mass (kilograms), fat mass (in kilograms and %) and fat free mass (in kilograms) were recorded from the assessment.

### Sarcopenia parameters

Skeletal muscle strength was evaluated by the hand-grip test, evaluated with a hand dynamometer (Yamar, Patterson Medical, UK). The participant performed the test with dominant hand in a seating position with the elbow flexed at 90 degrees and positioned on the side, but not against, the trunk. The hand was positioned firmly on the dynamometer with the thumb pointing up. The average of three trails measured in kilograms was considered for further analysis. Sarcopenia cut-off value for grip strength was defined by European Working Groups on Sarcopenia in Older People 2 (EWGSOP2) (*Cruz-Jentoft et al., 2019*), being <16 kg for women and <27 kg for men.

Skeletal muscle mass was evaluated using bioimpedance (see Anthropometric characteristics and body composition section). Moreover, the absolute level of skeletal muscle mass was adjusted for body height squared to obtain skeletal muscle index (SMI) (*Shepherd, 2016*). Cut-off points for SMI was identified using the algorithm of the European Working Groups on Sarcopenia in Older People (EWGSOP) (*Cruz-Jentoft et al., 2010*), $\geq 10.76$ kg/m$^2$ for men and $\geq 6.76$ kg/m$^2$ for women.

Physical performance was evaluated by determining gait speed and Timed up and go test (TUG). To evaluate gait speed, participants underwent a standardized assessment: they walked from side to side for 1 min and 30 s at their usual pace between 4 meters long system. Sarcopenia cut-off value for gait speed was defined by EWGSOP2 (*Cruz-Jentoft et al., 2019*), being <0.8 m/s for both sexes. TUG, as a valid measure of mobility, balance, and the ability to perform activities of daily living (*Beauchet et al., 2011*), was performed with participants seated in an armchair. Participants were measured for their ability to stand up from the chair, walk a 3-meter distance, turn around, walk back to the chair and sit down again (in seconds) (*Podsiadlo & Richardson, 1991*). Cut-off point for TUG was set, according to EWGSOP2(*Cruz-Jentoft et al., 2019*), at $\geq 20$ s.

### Participation in youth sports and socio-demographic questionnaire

We additionally obtained socio-demographic data with a self-reported questionnaire about prior involvement in sports in youth, comorbidities and number of medications. We were interested in participants' engagement in organized and competitive forms of sport in primary, secondary school and college and the number of years they participated in their

chosen sport. For that purpose, we divided them in two groups; participants who were engaged in organized forms of sport in their youth and therefore have athlete history (AH-athlete history) and participants who were not engaged in organized forms of sport in their youth and do not have athlete history (NAH-non athlete history). With the question "Were you active in sports as a child or adolescent, were you a member of a sports association or club, did you regularly participate in training sessions?" in the sociodemographic questionnaire participants were divided into two groups (AH or NAH). The additional questions "Which sport did you participate in the longest?" and "What were your greatest achievements in competitive sports?" helped to classify subjects according to their participation in competitive sports in their youth.

### Objectively measured physical activity level and sedentary behavior

The Actigraphy GT3x accelerometer (Actigraph; Pensacola, FL, USA) was used to assess adherence to the PA guideline (*Bull et al., 2020*) regarding 150 weekly minutes of MVPA. Furthermore, we assessed SB, too. Participants were instructed to wear the accelerometer around the right waist, attached by an elastic strap, during all waking hours (except for water-based activities) for a period of seven consecutive days (5 weekdays and 2 weekend days). The participants were verbally instructed to wear the accelerometer from the moment they woke up until bedtime at night. Also, they were asked to note in the diary provided the time when they wear the devices, when they were removed at the end of the day, and any time when the devices were removed and re-attached during the day. Accelerometers collected data at a frequency of 30 Hz and were converted into counts of movement in 10 s epochs. Continuous data was stored on a local memory chip and at the end of period transferred on a computer for further analysis. Data were processed using standard methods; all data outside the wearing time were removed, also data of 20 min of consecutive zeros were removed, too. Inclusion criteria for data validation were at least 10 h (>600 min) (*Choi et al., 2011*; *Hart et al., 2011*) of wearing time for a valid day and at least 5 days for a valid record, including one weekend day (*Mâsse et al., 2005*; *Trost, Mciver & Pate, 2005*). From a valid record, an overall physical activity (in cpm) was calculated. Furthermore, time spent in SB and MVPA was estimated using cpm-based thresholds: SB <100 cpm (*Lohne-Seiler et al., 2014*) and MVPA (>1,041 cpm) (*Copeland & Esliger, 2009*). A SB bout was characterized as an interval of at least 30 min of uninterrupted records of <100 cpm data (*Hernández-Vicente et al., 2019*), whereas sedentary break as characterized as at least 1 min of >100 cpm between two sedentary bouts. A MVPA bout was characterized as an interval of at least 10 min of uninterrupted records of >1,041 cpm data (*Gorman et al., 2014*).

### Statistical analyses

All statistical analyses were performed using IBM SPSS 21.0 statistics software (IBM Corp, Armonk, NY, USA). All anthropometric characteristics, and sarcopenic parameters are described using mean value, standard deviation, and minimal and maximal values. The normality of the distribution was checked and confirmed graphically (histogram and QQ plots) and statistically (the Shapiro–Wilk test) which was implemented to evaluate

**Table 1 Sample characteristics.**

| Characteristics | Mean ± SD (min–max) |
| --- | --- |
| Age (years) | 76.1 ± 5.2 (68.0–86.0) |
| Body height (cm) | 166.7 ± 9.1 (153.5–184.0) |
| Body mass (kg) | 73.0 ± 14.4 (48.3–106.5) |
| Body mass index (kg/m$^2$) | 26.1 ± 4.1 (19.6–40.8) |
| Fat free mass (kg) | 55.0 ± 11.9 (37.4–83.5) |
| Muscle mass (kg) | 24.9 ± 7.8 (13.1–43.9) |
| Fat mass (kg) | 17.9 ± 7.6 (5.8–44.3) |
| Fat mass (%) | 33.9 ± 6.5 (22.4–47.4) |
| Comorbidities (number/person) | 3.4 ± 2.0 (0–8) |
| Medications (number/person) | 1.8 ± 1.7 (0–6) |
| Sarcopenia parameters: | |
| Hand-grip strength (kg) | 32.3 ± 12.4 (12–60) |
| Skeletal muscle index (kg/m$^2$) | 8.8 ± 1.9 (5.5–13.0) |
| Gait speed (m/s) | 1.1 ± 0.2 (0.6–1.6) |
| Timed up and go test (sec) | 7.0 ± 1.7 (4.7–12.6) |

their normal distribution. We used t-test for independent samples to evaluate the possible differences between AH and NAH in MVPA or SB patterns. Further analysis of sarcopenia parameters was stratified by participant's sport participation in youth; to analyze the differences between groups we conducted analysis of covariance (ANCOVA), where age and sex were covariates. Multiple regression analysis was used to correlate SB and PA patterns to sarcopenia parameters, with age as a covariate. Significance was set at $\alpha \leq 5\%$.

## RESULTS

Of the 52 enrolled participants, we included in the analysis 44 participants, whose characteristics are described in Table 1. We excluded eight participants measurements from the analysis, due to refusal to wear an accelerometer ($N = 5$) or did not meet the inclusion criteria for data validation of accelerometer ($N = 3$). Average wear time of the accelerometers was 916.4 ± 98.2 min per day. Table 1 summarizes the anthropometric, health and sarcopenia parameters for the entire sample.

Table 2 provides descriptive data of PA, SB and athlete history in youth. Included participants were active older adults with average number 8,580 ± 3,536 of steps per day, they were engaged in MVPA on average 92.3 ± 45.5 min per day. Nevertheless, they were still highly sedentary, on average 10.9 ± 1.4 h per day. In addition, in the sample analyzed, 18 participants (out of 44) reported having participated in organized and competitive sports in their youth (mean age of participation 10.1 ± 6.7 years).

When we compared physical activity and sedentary behavior habits between AH and NAH, we found no differences. The AH group took on average of 8,106 ± 3,547 steps per day and was MVPA active 87.3 ± 45.3 min/day. The NAH group took 8,909 ± 3,561 steps and was MVPA active 95.7 ± 46.2 min/day ($p = 0.466$ and $p = 0.552$, respectively). We also

**Table 2 Physical and sport activity characteristics.**

| Characteristics | Mean ± SD (min–max) |
| --- | --- |
| Wearing time (min/day) | 916.4 ± 98.2 (702.0–1,140.0) |
| Overall physical activity (cpm) | 40.4 ± 12.4 (18.9–69.7) |
| Steps (number/day) | 8,580 ± 3.536 (2,671–16,796) |
| MVPA (min/day) | 92.3 ± 45.5 (19.6–187.7) |
| MVPA (bouts/day) | 5.0 ± 3.0 (1.0–13.0) |
| MVPA (mean bout time/day) | 52.4 ± 29.8 (10.0–130.3) |
| MVPA (min/week) | 599.7 ± 292.3 (117.8–1,345.8) |
| Sedentary behavior (min/day) | 656.2 ± 87.1 (463.3–913.8) |
| Sedentary behavior (no. bouts/day) | 2.3 ± 1.2 (0.8–5.7) |
| Sedentary behavior (mean bout time/day) | 95.3 ± 35.6 (33.0–203.2) |
| Athlete history | |
| Participation (N) | 18 (40.9%) |
| Participation (years) | 10.1 ± 6.7 (3–21) |

Note:
MVPA, Moderate to vigorous physical activity.

found no differences between the two selected groups at SB, with the AH group sitting an average of 675.5 ± 86.3 min/day and the NAH group sitting 642.7 ± 86.7 min/day ($p = 0.224$).

However, when we further analyzed the differences in sarcopenia parameters after dividing the sample into two groups according to sports participation in youth, we found differences in SMI and GRIP (Table 3). In addition, there were significant differences in GRIP and SMI between groups after controlling for age and not after controlling for sex and after controlling for sex and age. The differences pointed in favor of participants who had athlete history. In addition, we found no differences between SB or MVPA patterns, such as number of steps per day, mean MVPA bout time per day, number of MVPA bouts per day, total MVPA time per day, mean SB bout length per day, number of SB bouts per day, and total SB time per day.

Table 4 represents the associations between SB and sarcopenia parameters, after adjusting for age. There were associations between hand-grip strength and all included SB patterns (bout length, bout frequency and total time of SB per day) and TUG and all included SB patterns. SMI was associated with overall SB per day. Table 5 represents the associations between MVPA and sarcopenia parameters after adjusting for age. We found hand-grip strength, gait speed and TUG associated with all included MVPA parameters (such as number of steps per day, mean bout time per day, number of bouts per day and overall MVPA time per day).

## DISCUSSION

The aim of this study was to investigate active older adults with and without athlete history in youth. We were interested if active older adults who had athlete history also had different results on sarcopenia parameters in comparison to participants who did not have athlete history. Moreover, we were interested in regularly active older adults and their

**Table 3 Comparison of sarcopenia parameters between participation with athlete history in without athlete history, adjusted for age and sex.**

| Adjusted for | Sarcopenia parameters | AH<br>Mean ± SD | NAH<br>Mean ± SD | $p$-value<br>($\eta^2$) |
|---|---|---|---|---|
| Age | Hand-grip strength (kg) | 38.2 ± 2.5 | 28.1 ± 2.1 | 0.004 (0.185) |
| | Skeletal muscle index (kg/m$^2$) | 9.7 ± 0.4 | 8.2 ± 0.3 | 0.007 (0.165) |
| | Gait speed (m/s) | 1.1 ± 0.04 | 1.0 ± 0.04 | 0.363 (0.020) |
| | Timed up and go test (s) | 6.7 ± 0.4 | 7.2 ± 0.3 | 0.263 (0.031) |
| Sex | Hand-grip strength (kg) | 33.2 ± 1.9 | 31.6 ± 1.6 | 0.515 (0.010) |
| | Skeletal muscle index (kg/m$^2$) | 8.9 ± 0.3 | 8.7 ± 0.2 | 0.581 (0.007) |
| | Gait speed (m/s) | 1.1 ± 0.05 | 1.0 ± 0.04 | 0.624 (0.005) |
| | Timed up and go test (sec) | 6.8 ± 0.4 | 7.2 ± 0.3 | 0.557 (0.008) |
| Age × sex | Hand-grip strength (kg) | 34.2 ± 1.7 | 30.9 ± 1.4 | 0.162 (0.048) |
| | Skeletal muscle index (kg/m$^2$) | 9.0 ± 0.3 | 8.6 ± 0.2 | 0.286 (0.028) |
| | Gait speed (m/s) | 1.1 ± 0.05 | 1.0 ± 0.04 | 0.181 (0.044) |
| | Timed up and go test (sec) | 6.5 ± 0.4 | 7.3 ± 0.3 | 0.123 (0.059) |

**Note:**
AH, athlete histroy; NAH, non athlete history.

**Table 4 Multilinear regression of different sedentary behavior patterns, after adjusting for age, and sarcopenia parameters.**

| Parameters | B | β | 95% CI | R | R$^2$ | $p$-value |
|---|---|---|---|---|---|---|
| Hand-grip strength (kg) | | | | | | |
| SB bout length (min/day) | 0.010 | 0.028 | [−0.096 to 0.115] | 0.371 | 0.138 | 0.048 |
| SB bout frequency (bouts/day) | 1.186 | 0.112 | [−1.941 to 4.312] | 0.386 | 0.149 | 0.036 |
| SB (min/day) | 0.010 | 0.071 | [−0.033 to 0.053] | 0.377 | 0.142 | 0.043 |
| Skeletal muscle index (kg/m$^2$) | | | | | | |
| SB bout length (min/day) | 0.008 | 0.151 | [−0.009 to 0.025] | 0.330 | 0.109 | 0.094 |
| SB bout frequency (bouts/day) | 0.297 | 0.180 | [−0.200 to 0.795] | 0.345 | 0.119 | 0.075 |
| SB (min/day) | 0.006 | 0.288 | [0.000–0.013] | 0.408 | 0.166 | 0.024 |
| Gait speed (m/s) | | | | | | |
| SB bout (mean bout time/day) | −0.001 | −0.128 | [−0.003 to 0.001] | 0.493 | 0.243 | 0.003 |
| SB bout (no. bouts/day) | −0.038 | −0.207 | [−0.089 to 0.012] | 0.519 | 0.270 | 0.002 |
| SB (min/day) | −1.902E-5 | −0.008 | [−0.032 to −0.008] | 0.478 | 0.228 | 0.005 |
| Timed up and go test (sec) | | | | | | |
| SB bout (mean bout time/day) | 0.009 | 0.183 | [−0.004 to 0.022] | 0.531 | 0.282 | 0.001 |
| SB bout (no. bouts/day) | 0.243 | 0.165 | [−0.158 to 0.643] | 0.526 | 0.277 | 0.001 |
| SB (min/day) | 0.004 | 0.203 | [−0.001 to 0.009] | 0.538 | 0.289 | 0.001 |

**Note:**
SB, sedentary behavior.

relationship between SB, PA and selected sarcopenia parameters. Our research focus was on older adults who are physically active, a group that has been overlooked in studies on sarcopenia. However, because they make for an ideal study group, these individuals tend to be regularly engaged in PA, which allows us to examine how higher levels of PA or SB

**Table 5 Multilinear regression of different moderate-to-vigorous physical activity (MVPA) patterns, after adjusting for age, and sarcopenia parameters.**

| Parameters | B | β | 95% CI | R | $R^2$ | *p*-value |
|---|---|---|---|---|---|---|
| Hand-grip strength (kg) | | | | | | |
| STEPS (no./day) | 0.287E-3 | 0.082 | [−0.001 to 0.002] | 0.377 | 0.142 | 0.043 |
| MVPA bout (mean bout time/day) | 0.117 | 0.280 | [−0.015 to 0.248] | 0.447 | 0.200 | 0.010 |
| MVPA bout (no. bouts/day) | 1.320 | 0.316 | [0.028–2.612] | 0.467 | 0.218 | 0.006 |
| MVPA (min/day) | 0.014 | 0.051 | [−0.083 to 0.111] | 0.373 | 0.139 | 0.047 |
| Skeletal muscle index (kg/m$^2$) | | | | | | |
| STEPS (no./day) | −0.106E-3 | −0.193 | [−0.302E-3 to 0.900E-4] | 0.336 | 0.113 | 0.085 |
| MVPA bout (mean bout time/day) | 0.003 | 0.047 | [−0.019 to 0.025] | 0.299 | 0.089 | 0.147 |
| MVPA bout (no. bouts/day) | 0.065 | 0.099 | [−0.152 to 0.282] | 0.309 | 0.096 | 0.128 |
| MVPA (min/day) | −0.010 | −0.235 | [−0.025 to 0.005] | 0.353 | 0.125 | 0.065 |
| Gait speed (m/s) | | | | | | |
| STEPS (no./day) | −0.199E-5 | −0.032 | [−0.200E-5 to 0.180E-4] | 0.478 | 0.229 | 0.005 |
| MVPA bout (mean bout time/day) | −0.002 | −0.0291 | [−0.004 to 0.390E-4] | 0.544 | 0.296 | 0.001 |
| MVPA bout (no. bouts/day) | −0.023 | −0.308 | [−0.044 to 0.001] | 0.553 | 0.305 | 0.001 |
| MVPA (min/day) | −0.466E-3 | −0.098 | [−0.002 to 0.001] | 0.484 | 0.234 | 0.004 |
| Timed up and go test (s) | | | | | | |
| STEPS (no./day) | −0.9344E-4 | −0.192 | [−0.251E-3 to 0.640E-4] | 0.525 | 0.276 | 0.001 |
| MVPA bout (mean bout time/day) | −0.011 | −0.182 | [−0.028 to 0.007] | 0.526 | 0.277 | 0.001 |
| MVPA bout (no. bouts/day) | −0.089 | −0.154 | [−0.263 to 0.084] | 0.519 | 0.270 | 0.002 |
| MVPA (min/day) | −0.009 | −0.240 | [−0.021 to 0.003] | 0.538 | 0.289 | 0.001 |

**Note:**
MVPA, Moderate to vigorous physical activity.

might impact sarcopenia parameters. Moreover, the positive results could serve as an additional initiative for the development of public health guidelines and convince people to exercise more to protect their health.

When comparing participants with athlete history and without athlete history, and adjusted for age, we found differences in GRIP and SMI, which are two main sarcopenia parameters. The results are consistent with the only study we could find with a similar question-is there a difference between older adults who were engaged in sports in youth and participants who were not in sarcopenia parameters (*Tanaka et al., 2021*). On the contrary with published results, we found no differences between the groups in gait speed. The lack of literature on this subject raises a number of questions that need to be answered. It has already been proven that participation in sports activities at young age and continuation of these activities throughout a life, can have overall positive effect on the social, psychological, and physiological domains of ageing (*Göksu et al., 2019*; *Baker et al., 2010*; *Hirvensalo & Lintunen, 2011*). But can active participation in sport at a young age also have a positive impact on skeletal muscle disorders such as sarcopenia in old age? This may be an important message about the importance of participation in sports in youth and healthy ageing but needs to be analyzed in a larger sample. We were also interested in whether there was a difference in the present SB or PA patterns between participants with

athlete history (AH) and participants without athlete history (NAH) but found no difference. This data opens up new areas of research in the field of sarcopenia, as this could be of additional value in encouraging young people to participate in sports and invest in health for old age. Before this can happen, further research is needed to confirm or refute this issue.

Because we did not find differences in PA and SB habits between the AH and NAH groups, further investigation of sarcopenia parameters was performed on the entire sample. Physical inactivity has been linked to the onset of sarcopenia, with a low daily step count contributing to declining muscle strength and muscle mass (*Dent et al., 2018*; *Lee et al., 2018*; *Steffl et al., 2017*). On the other hand, higher levels of PA have been associated with a lower risk of sarcopenia (*Sánchez-Sánchez et al., 2019*; *Westbury et al., 2018*). Our sample included 44 older adults who overachieved recommended physical activity according to different recommendations; 42 of the 44 participants achieved the recommended levels of at least 150 min of PA per week (*Bull et al., 2020*), 27 of the 44 participants achieved the mean value of steps per day for the classification of somewhat active older adults (*Tudor-Locke et al., 2011*). Moreover, on average, participants had $5.0 \pm 3.0$ MVPA bouts per day, each one lasted on average $52.4 \pm 29.8$ min. We were interested in whether specific MVPA and SB patterns can be found in association with sarcopenic features. After adjusting for age, participants with a higher number of daily steps, longer MVPA bouts, a higher number of MVPA bouts in a day and higher overall MVPA engagement achieved better results in hand-grip strength and TUG. *Cooper et al. (2015)* already established that greater time spent in MVPA is positively associated with higher levels of physical capability, which can have an important impact on overall health and therefore on development of sarcopenia. Although we found a negative relationship between gait speed and MVPA, the regression coefficient was small. *Sánchez-Sánchez et al. (2019)* concluded that increased MVPA per day can have positive effects on gait speed in older adults, but that there is also a threshold at which improvements diminish. They suggested that more than 1.5 h of MVPA per day may have a ceiling effect. Since our subjects were active for an average of little more than 1.5 h per day, this could be why the curve began to turn negative.

We were also interested in participants' SB because of its negative effects on health in general and, consequently, on sarcopenia. There is limited information on sarcopenia parameters in active older adults, but because of the adverse effects already noted, further investigation is needed. Included participants sat an average of $656.2 \pm 87.1$ min per day and had on average $2.3 \pm 1.2$ SB bouts per day. The results on this topic are heterogeneous in the literature; on one hand, collected measurements are in accordance with literature that included the general population, which is usually less active (*Loyen et al., 2017*), but on the other hand researched reported lower values of SB (*Liao et al., 2018*; *Sánchez-Sánchez et al., 2019*). The research area has already recognized the importance of SB bouts and the lengths of SB bouts and their effect on health. Following the example of studies conducted by *Liao et al. (2018)*, *Wilson et al. (2021)* and *Hernández-Vicente et al. (2019)* we were

 

interested in, how different patterns of SB can affect sarcopenia parameters in active population. After adjusting for age, participants with fewer total SB, less SB bouts and shorter SB bouts had better TUG and gait speed results. The results are in accordance with the selected literature, but we must note that our sample still had high SB. In combination with high levels of PA, participants still had high SB; out of a total of 15.3 h per day of wearing time, the subjects spent 10.9 h in SB. Contrary to previously published studies (*Gardiner et al., 2011*; *Gianoudis, Bailey & Daly, 2015*), we found medium negative correlation between total SB and total MVPA, and therefore we could not use one variable as covariate in other's regression model.

Although this study investigated topical issues in a population that is not often represented in the literature, several limitations of the present study should be noted. Study limitations include small sample size compared to similar research. Therefore, the sample size is too small to draw firm conclusions, but it provides an indication of what needs to be explored further. Moreover, we could not establish a causal relationship PA or SB and sarcopenia parameters and therefore we could not make straight forward conclusions. In addition, a broader range of factors influencing age-related loss of muscle strength and muscle mass must be included in the analysis that will lead to definite conclusions. Although the accelerometer data are the most objective method for collecting physical activity habits, the data is limited in capturing postural information such as sitting *vs.* standing still. This can easily lead to overestimating sedentary time.

The important strength of the study is objectively measured PA and SB on active older adults, which could not be found in the literature. Therefore, this study provides an interesting perspective on the habits of older adults PA and SB and their influence on sarcopenia parameters—when comparing participants with athlete history and without athlete history. Additionally, the highlighted topic explored can trigger the upgrade of guidelines for PA and SB.

## CONCLUSIONS

Our findings suggest that engaging in sports activities in youth can make a difference with sarcopenia parameters. Although we found no differences in PA and SB habits between participants with AH and NAH, participants with an athlete history performed better results in sarcopenia parameters. We found significant differences in sarcopenia parameters between participants with and without athlete history, suggesting the importance of engaging in organized sport at a young age. Moreover, sedentary behavior and MVPA can be important factors for sarcopenia parameters such as hand grip strength, gait speed and TUG. Due to the aforementioned limitations of the study, these results need to be validated on a larger sample, which opens a new area of research on this topic.

## ACKNOWLEDGEMENTS

We would like to thank the participants of these measurements and the research team for their contributions that led to the success of the study.

### Funding

This research was conducted in the framework of the PANGeA project, CB147 Physical Activity and Nutrition for Quality Ageing, supported by the Cross-border Cooperation Program Slovenia–Italy 2007–2013 and cofinanced by European Regional Development Fund Grant 042-2/2009-18/052012. The research was funded by ARIS Research program, Grant Number P5-0381, Kinesiology for quality of life and as well as Slovenian national project L5-5550—Development of noninvasive marker for muscle atrophy (Grant No. 1000-15-1988), conducted by Institute for kinesiology research Science and Research Centre Koper Slovenia. The funders had no role in study design, data collection and analysis, decision to publish, or preparation of the manuscript.

### Grant Disclosures

The following grant information was disclosed by the authors:
Cross-Border Cooperation Program Slovenia–Italy: 2007–2013.
European Regional Development Fund Grant: 042-2/2009-18/052012.
ARIS Research Program: P5-0381.
Institute for kinesiology research Science and Research Centre Koper Slovenia: 1000-15-1988.

### Competing Interests

The authors declare that they have no competing interests.

### Author Contributions

- Kaja Teraž conceived and designed the experiments, performed the experiments, analyzed the data, prepared figures and/or tables, authored or reviewed drafts of the article, and approved the final draft.
- Miloš Kalc analyzed the data, authored or reviewed drafts of the article, and approved the final draft.
- Boštjan Šimunič conceived and designed the experiments, performed the experiments, prepared figures and/or tables, authored or reviewed drafts of the article, and approved the final draft.
- Uros Marusic analyzed the data, authored or reviewed drafts of the article, and approved the final draft.
- Primož Pori conceived and designed the experiments, authored or reviewed drafts of the article, and approved the final draft.
- Saša Pišot conceived and designed the experiments, performed the experiments, analyzed the data, authored or reviewed drafts of the article, and approved the final draft.
- Rado Pišot conceived and designed the experiments, authored or reviewed drafts of the article, and approved the final draft.

## Human Ethics

The following information was supplied relating to ethical approvals (*i.e.*, approving body and any reference numbers):

National Ethical Committee of Slovenian Ministry of Health (ethical approval no. 0120-76/2021/6) and confirmed by the ZRS Koper Scientific Council no. 0624-77/21.

## Data Availability

The raw data are available in the Supplemental File.

## Supplemental Information

Supplemental information for this article can be found online at http://dx.doi.org/10.7717/peerj.16432#supplemental-information.

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
