# Peer review of "Participation in youth sports influences sarcopenia parameters in older adults"

_PeerJ, doi:10.7717/peerj.16432_

## Round 0.1 · original submission · Major Revisions

Thank you very much for submitting the manuscript entitled: "Participation in youth sports influences sarcopenia parameters in older adults" to PeerJ.

Please pay attention to the recommendations and comments of the reviewers. There are some issues that must be resolved to improve the final manuscript before making the decision to publish it.

Receive a cordial greeting.

·

Basic reporting

Clear, unambiguous, professional English language is used throughout the article.
Literature references, sufficient field background/context provided.
The article includes sufficient introduction and background to demonstrate how the work fits into the broader field of knowledge and relevant prior literature is referenced.
The structure of the article conforms to an acceptable format of ‘standard sections’.
The figures generally correspond to the content of the article with sufficient resolution, and are appropriately described and labeled, but additional information about the respondents' prior involvement in sports at a younger age and physical activity and sedentary behavior between the AH-athlete history and NAH-no-athlete history groups should be added.
More appropriate raw data should be made available in accordance with Data Sharing policy.
The paper is self-contained with relevant results and hypotheses.
Coherent bodies of the article are appropriately subdivided.

Experimental design

The article describes original primary research within Aims and Scope of the journal.
The research question is well defined, relevant and meaningful. It is also stated how research fills an identified knowledge gap.
The article uses rigorous investigation performed to a high technical and ethical standard.
Methods described with sufficient detail and information to replicate by another investigator.

Validity of the findings

Impact and novelty of the research is assessed. Meaningful replication encouraged where rationale and benefit to literature is clearly stated.
Data are statistically controlled and they are robust, but some additional data should be provided.
Conclusions are well stated, linked to original research question and limited to supporting results.

Additional comments

The results of socio-demographic data of self-reported questionnaire about prior involvement in sports of respondents in younger age, should be described more properly. It would be necessary to indicate exactly, by what criteria the respondents were divided into two groups: participants who were engaged in organized sports in their youth and therefore have a history of being an athlete (AH-athlete history) and participants who did not engage in organized sports in their youth and have no history of being an athlete (NAH -non-athlete history).
It would be advisable to add more data (or prepare a table) on compared physical activity and sedentary behavior habits between AH-athlete history and NAH-non-athlete history groups.
It is recommended to supplement the description of the results of 4th and 5th tables.

Reviewer 2 ·

Basic reporting

The manuscript demonstrates clear, unambiguous, and professional use of the English language throughout. While not strictly necessary, I would make reference in the abstract to the number of women in the total sample, or alternatively, to the number of men and women. The introduction and background effectively establish the context. The literature references are well-cited and relevant. The manuscript's structure aligns with PeerJ standards.
Main review files, Tables 1, 2, 3, 4, and 5, are pertinent, of high quality, appropriately labeled, and well-described. However, in the Supplemental file: “English translation of the questionnaire” the translation is missing in question number 19
The authors have adhered to PeerJ's data supply policy. However, it is recommended that the Excel file include a first row with descriptive metadata to elucidate the subsequent data columns.

Experimental design

EXPERIMENTAL DESIGN
The research represents original primary research well within the scope of the journal. The research question is clearly defined, relevant, and meaningful. The manuscript effectively articulates how the research addresses an identified knowledge gap.
The investigation demonstrates rigor, maintaining high technical and ethical standards. The authors' ethical approval statement has been scrutinized and found appropriate. There are no unethical or unnecessary experiments.
The methods are detailed and provide sufficient information for replication. However, it is worth noting that the study employs a relatively small sample size, a limitation that does not preclude acceptance and would justify the conduct of new studies.

Validity of the findings

All underlying data have been supplied and exhibit robustness, statistical soundness, and control. The conclusions are well-stated and directly tied to the original research question, remaining within the confines of supporting the results.

Main Issues:
1. While the raw data has been graciously provided, it is recommended that the supplemental files include more descriptive metadata identifiers for the benefit of future readers. Specifically, the Excel file could benefit from a first row containing descriptive metadata that elucidates the subsequent data columns.
2. In the English translation of the questionnaire, translation of question number 19 is missing.
3. While not strictly necessary, I would make reference in the abstract to the number of women in the total sample, or alternatively, to the number of men and women.
4. Regarding language and grammar, while the overall writing is correct, there is room for improvement in some instances. Simplifying phrases and avoiding word repetition can enhance readability and comprehension. Examples where language can be refined include lines 145, 151, 152, 155-158.
5. Strengths and weaknesses of the manuscript: The authors are commended for their extensive data set, diligently collected during detailed fieldwork. Additionally, the manuscript is composed in a clear, professional, and unambiguous manner. If any weakness exists in the study, it lies in the sample size. However, this limitation is not incompatible with acceptance and may indeed justify the pursuit of further research.

---

## Round 0.2 · Minor Revisions

Dear Authors:

Sorry for the delay. Thank you for your patience.

Please, have a look at some minor reviews.

Regards

·

Basic reporting

Clear, unambiguous, professional technically correct English language is used throughout the article. Literature references and sufficient field background/context is provided.
The article includes sufficient introduction and background to demonstrate how the work fits into the broader field of knowledge and relevant prior literature is referenced.
The structure of the article conforms to an acceptable format of ‘standard sections’.
The figures generally correspond to the content of the article with sufficient resolution, and are appropriately described and labeled.
The paper is self-contained with relevant results and hypotheses.
Coherent bodies of the article are appropriately subdivided.

Experimental design

The article describes original primary research within Aims and Scope of the journal.
The research question is well defined, relevant and meaningful. It is also stated how research fills an identified knowledge gap.
The article uses rigorous investigation performed to a high technical and ethical standard.
Methods described with sufficient detail and information to replicate by another investigator.

Validity of the findings

Impact and novelty of the research is assessed. Meaningful replication encouraged where rationale and benefit to literature is clearly stated.
Data are statistically controlled and they are robust.
Conclusions are well stated, linked to original research question and limited to supporting results.

Additional comments

The previously noted flaws in the article have been corrected. The description of the results of socio-demographic data of self-reported questionnaire about prior involvement in sports of respondents in younger has been improved. Additional data on compared physical activity and sedentary behavior habits between AH-athlete history and NAH-non-athlete history groups have been added.
In the line 174 correct English must be used: … in the sociodemographic questionnaire participants were divided into two groups…

Reviewer 2 ·

Basic reporting

no comment

Experimental design

no comment

Validity of the findings

no comment

Additional comments

language can be refined in lines 155-158.
In lines 155-158, what I propose is to make the text simpler. To be a scientific text, the sentence is too long, and it could be divided into two simpler sentences.

---

## Round 0.3 · accepted · Accept

Dear co-authors:

I am pleased to tell you that your manuscript has been accepted for publication in PeerJ. We appreciate you considering PeerJ to publish your research work and for your patience. We have had difficulties finding suitable reviewers, which has delayed the final decision a bit.

Congratulations and best regards.

Dr. Manuel Jiménez

Reviewer 2 ·

Basic reporting

no comment

Experimental design

no comment

Validity of the findings

no comment

Additional comments

no comment.